# Bayesian Learning of Sum-Product Networks

**Martin Trapp**[1,2], **Robert Peharz**[3], **Hong Ge**[3],
**Franz Pernkopf**[1], **Zoubin Ghahramani**[4,3]
[1]Graz University of Technology, [2]OFAI,
[3]University of Cambridge, [4]Uber AI
martin.trapp@tugraz.at, rp587@cam.ac.uk, hg344@cam.ac.uk
pernkopf@tugraz.at, zoubin@eng.cam.ac.uk

## Abstract

Sum-product networks (SPNs) are flexible density estimators and have received significant attention due to their attractive inference properties. While parameter learning in SPNs is well developed, structure learning leaves something to be desired: Even though there is a plethora of SPN structure learners, most of them are somewhat ad-hoc and based on intuition rather than a clear learning principle. In this paper, we introduce a well-principled Bayesian framework for SPN structure learning. First, we decompose the problem into i) laying out a computational graph, and ii) learning the so-called scope function over the graph. The first is rather unproblematic and akin to neural network architecture validation. The second represents the effective structure of the SPN and needs to respect the usual structural constraints in SPN, i.e. completeness and decomposability. While representing and learning the scope function is somewhat involved in general, in this paper, we propose a natural parametrisation for an important and widely used special case of SPNs. These structural parameters are incorporated into a Bayesian model, such that simultaneous structure and parameter learning is cast into monolithic Bayesian posterior inference. In various experiments, our Bayesian SPNs often improve test likelihoods over greedy SPN learners. Further, since the Bayesian framework protects against overfitting, we can evaluate hyper-parameters directly on the Bayesian model score, waiving the need for a separate validation set, which is especially beneficial in low data regimes. Bayesian SPNs can be applied to heterogeneous domains and can easily be extended to nonparametric formulations. Moreover, our Bayesian approach is the first, which consistently and robustly learns SPN structures under missing data.

## 1 Introduction

Sum-product networks (SPNs) [29] are a prominent type of deep probabilistic model, as they are a flexible representation for high-dimensional distributions, yet allowing fast and exact inference. Learning SPNs can be naturally organised into *structure learning* and *parameter learning*, following the same dichotomy as in probabilistic graphical models (PGMs) [16]. Like in PGMs, state-of-the-art SPN parameter learning covers a wide range of well-developed techniques. In particular, various *maximum likelihood* approaches have been proposed using either gradient-based optimisation [37, 27, 3, 40] or expectation-maximisation (and related schemes) [25, 29, 46]. In addition, several *discriminative* criteria, e.g. [9, 14, 39, 32], as well as Bayesian approaches to parameter learning, e.g. [44, 33, 43], have been developed.

Concerning *structure learning*, however, the situation is remarkably different. Although there is a plethora of structure learning approaches for SPNs, most of them can be described as a heuristic. For example, the most prominent structure learning scheme, LearnSPN [10], derives an SPN structure

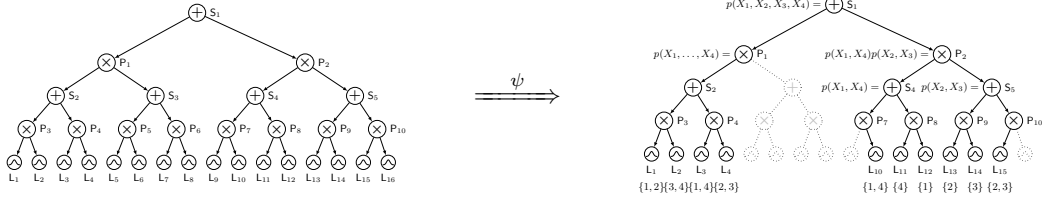

Figure 1: A computational graph $\mathcal{G}$ (left) and an SPN structure (right), defined by the scope function $\psi$, discovered using posterior inference on an encoding of $\psi$. The SPN contains only a subset of the nodes in $\mathcal{G}$ as some sub-trees are allocated with an empty scope (dotted) – evaluating to constant 1. Note that the graph $\mathcal{G}$ only encodes the topological layout of nodes, while the "effective" SPN structure is encoded via $\psi$.

by recursively clustering the data instances (yielding sum nodes) and partitioning data dimensions (yielding product nodes). Each of these steps can be understood as some local structure improvement, and as an attempt to optimise a local criterion. While LearnSPN is an intuitive scheme and elegantly maps the structural SPN semantics onto an algorithmic procedure, the fact that the *global goal of structure learning is not declared* is unsatisfying. This principal shortcoming of LearnSPN is shared by its many variants such as online LearnSPN [18], ID-SPN [35], LearnSPN-b [42], mixed SPNs [22], and automatic Bayesian density analysis (ABDA) [43]. Also other approaches lack a sound learning principle, such as [5, 1] which derive SPN structures from k-means and SVD clustering, respectively, [23] which grows SPNs bottom up using a heuristic based on the information bottleneck, [6] which uses a heuristic structure exploration, or [13] which use a variant of hard EM to decide when to enlarge or shrink an SPN structure.

All of the approaches mentioned above fall short of posing some fundamental questions: *What is a good SPN structure?* or *What is a good principle to derive an SPN structure?* This situation is somewhat surprising since the literature on PGMs offers a rich set of learning principles: In PGMs, the main strategy is to optimise a *structure score* such as minimum-description-length (MDL) [38], Bayesian information criterion (BIC) [16] or the Bayes-Dirichlet (BD) score [4, 11]. Moreover, in [7] an approximate MCMC sampler was proposed for full Bayesian structure learning.

In this paper, we propose a well-principled Bayesian approach to SPN learning, by simultaneously performing inference over both structure and parameters. We first decompose the structure learning problem into two steps, namely i) proposing a *computational graph*, laying out the arrangement of sums, products and leaf distributions, and ii) learning the so-called *scope-function*, which assigns to each node its scope.[1] The first step is straightforward, computational graphs have only very few requirements, while the second step, learning the scope function, is more involved in full generality. Therefore, we propose a parametrisation of the scope function for a widely used special case of SPNs, namely so-called *tree-shaped region graphs* [5, 27]. This restriction allows us to encode the scope function via categorical variables elegantly. Now Bayesian learning becomes conceptually simple: We equip all latent variables and the leaves with appropriate priors and perform monolithic Bayesian inference, implemented via Gibbs-updates. Figure 1 illustrates our approach of disentangling structure learning and performing Bayesian inference on an encoding of the scope function.

In summary, our main contributions in this paper are:

- We propose a novel and well-principled approach to SPN structure learning, by decomposing the problem into finding a *computational graph* and learning a *scope-function*.

- To learn the scope function, we propose a natural parametrisation for an important sub-type of SPNs, which allows us to formulate a joint Bayesian framework simultaneously over structure and parameters.

- Bayesian SPNs are protected against overfitting, waiving the necessity of a separate validation set, which is beneficial for low data regimes. Furthermore, they naturally deal with missing data and are the first – to the best of our knowledge – which consistently and robustly learn SPN structures under missing data. Bayesian SPNs can easily be extended to nonparametric formulations, supporting growing data domains.

## 2   Related Work

The majority of structure learning approaches for SPNs, such as LearnSPN [10] and its variants, e.g. [18, 35, 42, 22] or other approaches, such as [5, 23, 1, 6, 13], heuristically generate a structure by optimising some local criterion. However, none of these approaches defines an overall goal of structure learning, and all of them lack a sound objective to derive SPN structures. Bayesian SPNs, on the other hand, follow a well-principled approach using posterior inference over structure and parameters.

The most notable attempts for principled structure learning of SPNs include ABDA [43] and the existing nonparametric variants of SPNs [17, 41]. Even though the Bayesian treatment in ABDA, i.e. posterior inference over the parameters of the latent variable models located at the leaf nodes, can be understood as some kind of local Bayesian structure learning, the approach heavily relies on a heuristically predefined SPN structure. In fact, Bayesian inference in ABDA only allows adaptations of the parameters and the latent variables at the leaves and does not infer the general structure of the SPN. Therefore, ABDA can be understood as a particular case of Bayesian SPNs in which the overall structure is kept fixed, and inference is only performed over the latent variables at the leaves and the parameters of the SPN.

On the other hand, nonparametric formulations for SPNs, i.e. [17, 41], use Bayesian nonparametric priors for both structure and parameters. However, the existing approaches do not use an efficient representation, e.g. [41] uses uniform distributions over all possible partitions of the scope at each product node, making posterior inference infeasible for real-world applications. Bayesian SPNs and nonparametric extension of Bayesian SPNs, on the other hand, can be applied to real-world applications, as shown in Section 6.

Besides structure learning in SPNs, there are various approaches for other tractable probabilistic models (which also allow exact and efficient inference), such as probabilistic sentential decision diagrams (PSDDs) [15] and Cutset networks (CNets) [31]. Most notably, the work by [19] introduces a greedy approach to optimises a heuristically defined global objective for structure learning in PSDDs. However, similar to structure learning of selective SPNs [24] (which are a restricted sub-type of SPNs), the global objective of the optimisation is not well-principled. Existing approaches for CNets, on the other hand, mainly use heuristics to define the structure. In [21] structures are constructed randomly, while [30] compiles a learned latent variable model, such as an SPNs, into a CNet. However, all these approaches lack a sound objective which defines a good structure.

## 3   Background

Let $\mathbf{X} = \{X_1, \ldots, X_D\}$ be a set of $D$ random variables (RVs), for which $N$ i.i.d. samples are available. Let $x_{n,d}$ be the $n^{\text{th}}$ observation for the $d^{\text{th}}$ dimension and $\mathbf{x}_n := (x_{n,1}, \ldots, x_{n,D})$. Our goal is to estimate the distribution of $\mathbf{X}$ using a sum-product network (SPN). In the following we review SPNs, but use a more general definition than usual, in order to facilitate our discussion below. In this paper, we define an SPN $\mathcal{S}$ as a 4-tuple $\mathcal{S} = (\mathcal{G}, \psi, \mathbf{w}, \theta)$, where $\mathcal{G}$ is a *computational graph*, $\psi$ is a *scope-function*, $\mathbf{w}$ is a set of sum-weights, and $\theta$ is a set of leaf parameters. In the following, we explain these terms in more detail.

**Definition 1** (Computational graph). *The computational graph $\mathcal{G}$ is a connected directed acyclic graph, containing three types of nodes: sums (S), products (P) and leaves (L). A node in $\mathcal{G}$ has no children if and only if it is of type L. When we do not discriminate between node types, we use N for a generic node. S, P, L, and N denote the collections of all S, all P, all L, and all N in $\mathcal{G}$, respectively. The set of children of node N is denoted as $\mathbf{ch}(N)$. In this paper, we require that $\mathcal{G}$ has only a single root (node without parent).*

**Definition 2** (Scope function). *The scope function is a function $\psi \colon N \mapsto 2^{\mathbf{X}}$, assigning each node in $\mathcal{G}$ a sub-set of $\mathbf{X}$ ($2^{\mathbf{X}}$ denotes the power set of $\mathbf{X}$). It has the following properties:*

1. *If N is the root node, then $\psi(N) = \mathbf{X}$.*

2. *If N is a sum or product, then $\psi(N) = \bigcup_{N' \in \mathbf{ch}(N)} \psi(N')$.*

3. *For each $S \in \mathbf{S}$ we have $\forall N, N' \in \mathbf{ch}(S) \colon \psi(N) = \psi(N')$ (completeness).*

4. *For each $P \in \mathbf{P}$ we have $\forall N, N' \in \mathbf{ch}(P) \colon \psi(N) \cap \psi(N') = \emptyset$ (decomposability).*

Each node $N$ in $\mathcal{G}$ represents a distribution over the random variables $\psi(N)$, as described in the following. Each leaf $L$ computes a pre-specified distribution over its scope $\psi(L)$ (for $\psi(L) = \emptyset$, we set $L \equiv 1$). We assume that $L$ is parametrised by $\theta_L$, and that $\theta_L$ represents a distribution *for any possible choice of* $\psi(L)$. In the most naive setting, we would maintain a separate parameter set for each of the $2^D$ possible choices for $\psi(L)$, but this would quickly become intractable. In this paper, we simply assume that $\theta_L$ contains $D$ parameters $\theta_{L,1}, \ldots, \theta_{L,D}$ over single-dimensional distributions (e.g. Gaussian, Bernoulli, etc), and that for a given $\psi(L)$, the represented distribution factorises: $L = \prod_{X_i \in \psi(L)} p(X_i \mid \theta_{L,i})$. However, more elaborate schemes are possible. Note that our definition of leaves is quite distinct from prior art: previously, leaves were defined to be distributions over a fixed scope; our leaves define at all times distributions over *all* $2^D$ possible scopes. The set $\theta = \{\theta_L\}$ denotes the collection of parameters for all leaf nodes. A sum node $S$ computes a weighted sum $S = \sum_{N \in \mathbf{ch}(S)} w_{S,N} N$. Each weight $w_{S,N}$ is non-negative, and can w.l.o.g. [26, 45] be assumed to be normalised: $w_{S,N} \geq 0$, $\sum_{N \in \mathbf{ch}(S)} w_{S,N} = 1$. We denote the set of all sum-weights for $S$ as $\mathbf{w}_S$ and use $\mathbf{w}$ to denote the set of all sum-weights in the SPN. A product node $P$ simply computes $P = \prod_{N \in \mathbf{ch}(P)} N$.

The two conditions we require for $\psi$ – completeness and decomposability – ensure that each node $N$ is a probability distribution over $\psi(N)$. The distribution represented by $\mathcal{S}$ is defined as the distribution of the root node in $\mathcal{G}$ and denoted as $\mathcal{S}(\mathbf{x})$. Furthermore, completeness and decomposability are essential to render many inference scenarios tractable in SPNs. In particular, arbitrary marginalisation tasks reduce to marginalisation at the leaves, i.e. simplify to several marginalisation tasks over (small) subsets of $\mathbf{X}$, followed by an evaluation of the internal part (sum and products) in a simple feed-forward pass [26]. Thus, *exact* marginalisation can be computed in *linear time* in size of the SPN (assuming constant time marginalisation at the leaves). Conditioning can be tackled similarly. Note that marginalisation and conditioning are key inference routines in probabilistic reasoning so that SPNs are generally referred to as tractable probabilistic models.

# 4 Bayesian Sum-Product Networks

All previous work define the structure of an SPN in an entangled way, i.e. the scope is seen as an inherent property of the nodes in the graph. In this paper, however, we propose to decouple the aspects of an SPN structure by searching over $\mathcal{G}$ and nested learning of $\psi$. Note that $\mathcal{G}$ has few structural requirements, and can be validated like a neural network structure. Consequently, we fix $\mathcal{G}$ in the following discussion and cross-validate it in our experiments. Learning $\psi$ is challenging, as $\psi$ has non-trivial structure due to the completeness and decomposability conditions. In the following, we develop a parametrisation of $\psi$ and incorporate it into a Bayesian framework. We first revisit Bayesian parameter learning in SPNs using a fixed $\psi$.

## 4.1 Learning Parameters $\mathbf{w}, \theta$ – Fixing Scope Function $\psi$

The key insight for Bayesian parameter learning [44, 33, 43] is that sum nodes can be interpreted as latent variables, clustering data instances [29, 45, 25]. Formally, consider any sum node $S$ and assume that it has $K_S$ children. For each data instance $\mathbf{x}_n$ and each $S$, we introduce a latent variable $Z_{S,n}$ with $K_S$ states and categorical distribution given by the weights $\mathbf{w}_S$ of $S$. Intuitively, the sum node $S$ represents a latent clustering of data instances over its children. Let $\mathbf{Z}_n = \{Z_{S,n}\}_{S \in \mathbf{s}}$ be the collection of all $Z_{S,n}$. To establish the interpretation of sum nodes as latent variables, we introduce the notion of induced tree [44]. We omit sub-script $n$ when a distinction between data instances is not necessary.

**Definition 3** (Induced tree [44]). *Let an $\mathcal{S} = (\mathcal{G}, \psi, \mathbf{w}, \theta)$ be given. Consider a sub-graph $\mathcal{T}$ of $\mathcal{G}$ obtained as follows: i) for each sum $S \in \mathcal{G}$, delete all but one outgoing edge and ii) delete all nodes and edges which are now unreachable from the root. Any such $\mathcal{T}$ is called an* induced tree *of $\mathcal{G}$ (sometimes also denoted as induced tree of $\mathcal{S}$). The SPN distribution can always be written as the mixture*

$$\mathcal{S}(\mathbf{x}) = \sum_{\mathcal{T} \sim \mathcal{S}} \prod_{(S,N) \in \mathcal{T}} w_{S,N} \prod_{L \in \mathcal{T}} L(\mathbf{x}_L), \tag{1}$$

*where the sum runs over all possible induced trees in $\mathcal{S}$, and $L(\mathbf{x}_L)$ denotes the evaluation of $L$ on the restriction of $\mathbf{x}$ to $\psi(L)$.*

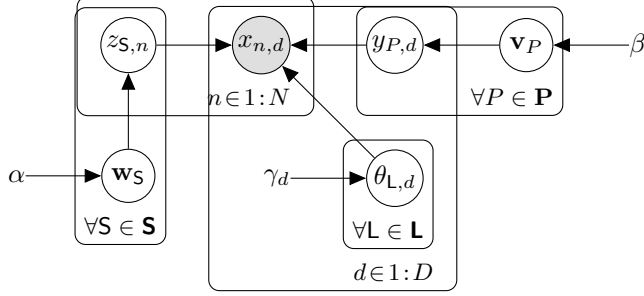

Figure 2: Plate notation of our generative model for Bayesian structure and parameter learning.

We define a function $T(\mathbf{z})$ which assigns to each value $\mathbf{z}$ of $\mathbf{Z}$ the *induced tree determined by* $\mathbf{z}$, i.e. where $\mathbf{z}$ indicates the kept sum edges in Definition 3. Note that the function $T(\mathbf{z})$ is surjective, but not injective, and thus, $T(\mathbf{z})$ is not invertible. However, it is "partially" invertible, in the following sense: Note that any $\mathcal{T}$ splits the set of all sum nodes $\mathbf{S}$ into two sets, namely the set of sum nodes $\mathbf{S}_{\mathcal{T}}$ which are contained in $\mathcal{T}$, and the set of sum nodes $\bar{\mathbf{S}}_{\mathcal{T}}$ which are not. For any $\mathcal{T}$, we can identify (invert) the state $z_{\mathsf{S}}$ for any $\mathsf{S} \in \mathbf{S}_{\mathcal{T}}$, as it corresponds to the unique child of $\mathsf{S}$ in $\mathcal{T}$. On the other hand, the state of any $\mathsf{S} \in \bar{\mathbf{S}}_{\mathcal{T}}$ is arbitrary. In short, given an induced tree $\mathcal{T}$, we can perfectly retrieve the states of the (latent variables of) sum nodes in $\mathcal{T}$, while the states of the other latent variables are arbitrary.

Now, define the conditional distribution $p(\mathbf{x} \mid \mathbf{z}) = \prod_{\mathsf{L} \in T(\mathbf{z})} L(\mathbf{x}_{\mathsf{L}})$ and prior $p(\mathbf{z}) = \prod_{\mathsf{S} \in \mathcal{G}} w_{\mathsf{S}, z_{\mathsf{S}}}$, where $w_{\mathsf{S}, z_{\mathsf{S}}}$ is the sum-weight indicated by $z_{\mathsf{S}}$. When marginalising $\mathbf{Z}$ from the joint $p(\mathbf{x}, \mathbf{z}) = p(\mathbf{x} \mid \mathbf{z}) \, p(\mathbf{z})$, we yield

$$\sum_{\mathbf{z}} \prod_{\mathsf{S} \in \mathbf{S}} w_{\mathsf{S}, z_{\mathsf{S}}} \prod_{\mathsf{L} \in T(\mathbf{z})} L(\mathbf{x}_{\mathsf{L}}) = \sum_{\mathcal{T}} \sum_{\mathbf{z} \in T^{-1}(\mathcal{T})} \prod_{\mathsf{S} \in \mathbf{S}} w_{\mathsf{S}, z_{\mathsf{S}}} \prod_{\mathsf{L} \in T(\mathbf{z})} L(\mathbf{x}_{\mathsf{L}}) \tag{2}$$

$$= \sum_{\mathcal{T}} \prod_{(\mathsf{S}, \mathsf{N}) \in \mathcal{T}} w_{\mathsf{S}, \mathsf{N}} \prod_{\mathsf{L} \in \mathcal{T}} \mathsf{L}(\mathbf{x}_{\mathsf{L}}) \underbrace{\left( \sum_{\bar{\mathbf{z}}} \prod_{\mathsf{S} \in \bar{\mathbf{S}}_{\mathcal{T}}} w_{\mathsf{S}, \bar{z}_{\mathsf{S}}} \right)}_{=1} = \mathcal{S}(\mathbf{x}), \tag{3}$$

establishing the SPN distribution (1) as latent variable model, with $\mathbf{Z}$ marginalised out. In (2), we split the sum over all $\mathbf{z}$ into the double sum over all induced trees $\mathcal{T}$, and all $\mathbf{z} \in T^{-1}(\mathcal{T})$, where $T^{-1}(\mathcal{T})$ is the pre-image of $\mathcal{T}$ under $T$, i.e. the set of all $\mathbf{z}$ for which $T(\mathbf{z}) = \mathcal{T}$. As discussed above, the set $T^{-1}(\mathcal{T})$ is made up by a unique z-assignment for each $\mathsf{S} \in \mathbf{S}_{\mathcal{T}}$, corresponding to the unique sum-edge $(\mathsf{S}, \mathsf{N}) \in \mathcal{T}$, and all possible assignments for $\mathsf{S} \in \bar{\mathbf{S}}_{\mathcal{T}}$, leading to (3).

It is now conceptually straightforward to extend the model to a Bayesian setting, by equipping the sum-weights $\mathbf{w}$ and leaf-parameters $\theta$ with suitable priors. In this paper, we assume Dirichlet priors for sum-weights and some parametric form $L(\cdot \mid \theta_{\mathsf{L}})$ for each leaf, with conjugate prior over $\theta_{\mathsf{L}}$, leading to the following generative model:

$$\mathbf{w}_{\mathsf{S}} \mid \alpha \sim \mathcal{D}ir(\mathbf{w}_{\mathsf{S}} \mid \alpha) \ \ \forall \mathsf{S}, \quad z_{\mathsf{S},n} \mid \mathbf{w}_{\mathsf{S}} \sim \mathcal{C}at(z_{\mathsf{S},n} \mid \mathbf{w}_{\mathsf{S}}) \ \ \forall \mathsf{S} \, \forall n,$$

$$\theta_{\mathsf{L}} \mid \gamma \sim p(\theta_{\mathsf{L}} \mid \gamma) \ \ \forall \mathsf{L}, \qquad \mathbf{x}_n \mid \mathbf{z}_n, \theta \sim \prod_{\mathsf{L} \in T(\mathbf{z}_n)} \mathsf{L}(\mathbf{x}_{\mathsf{L},n} \mid \theta_{\mathsf{L}}) \ \ \forall n. \tag{4}$$

We now extend the model to also comprise the SPN's "effective" structure, the scope function $\psi$.

### 4.2 Jointly Learning $\mathbf{w}$, $\theta$ and $\psi$

Given a computational graph $\mathcal{G}$, we wish to learn $\psi$, additionally to the SPN's parameters $\mathbf{w}$ and $\theta$, and adopt it in our generative model (4). In general graphs $\mathcal{G}$, representing $\psi$ in an amenable form is rather involved. Therefore, in this paper, we restrict to the class of SPNs whose computational $\mathcal{G}$ follows a *tree-shaped region graph*, which leads to a natural encoding of $\psi$. Region graphs can be understood as a "vectorised" representation of SPNs, and have been used in several SPN learners e.g. [5, 23, 27].

**Definition 4** (Region graph). *Given a set of random variables* $\mathbf{X}$, *a* region graph *is a tuple* $(\mathcal{R}, \psi)$ *where* $\mathcal{R}$ *is a connected directed acyclic graph containing two types of nodes: regions* $(R)$ *and partitions* $(P)$. $\mathcal{R}$ *is bipartite w.r.t. to these two types of nodes, i.e. children of* $R$ *are only of type* $P$ *and vice versa.* $\mathcal{R}$ *has a single root (node with no parents) of type* $R$, *and all leaves are also of type* $R$. *Let* $\mathbf{R}$ *be the set of all* $R$ *and* $\mathbf{P}$ *be the set of all* $P$. *The* scope function *is a function* $\psi\colon \mathbf{R} \cup \mathbf{P} \mapsto 2^{\mathbf{X}}$, *with the following properties: 1) If* $R \in \mathbf{R}$ *is the root, then* $\psi(R) = \mathbf{X}$. *2) If* $Q$ *is either a region with children or a partition, then* $\psi(Q) = \bigcup_{Q' \in \mathbf{ch}(Q)} \psi(Q')$. *3) For all* $P \in \mathbf{P}$ *we have* $\forall R, R' \in \mathbf{ch}(P)\colon \psi(R) \cap \psi(R') = \emptyset$. *4) For all* $R \in \mathbf{R}$ *we have* $\forall P \in \mathbf{ch}(R)\colon \psi(R) = \psi(P)$.

Note that, we generalised previous notions of a region graph [5, 23, 27], also decoupling its graphical structure $\mathcal{R}$ and the scope function $\psi$ (we are deliberately overloading symbol $\psi$ here). Given a region graph $(\mathcal{R}, \psi)$, we can easily construct an SPN structure $(\mathcal{G}, \psi)$ as follows. To construct the SPN graph $\mathcal{G}$, we introduce a single sum node for the root region in $\mathcal{R}$; this sum node will be the output of the SPN. For each leaf region $R$, we introduce $I$ SPN leaves. For each other region $R$, which is neither root nor leaf, we introduce $J$ sum nodes. Both $I$ and $J$ are hyper-parameters of the model. For each partition $P$ we introduce all possible cross-products of nodes from $P$'s child regions. More precisely, let $\mathbf{ch}(P) = \{R_1, \ldots R_K\}$. Let $\mathbf{N}_k$ be the assigned sets of nodes in each child region $R_k$. Now, we construct all possible cross-products $\mathsf{P} = \mathsf{N}_1 \times \cdots \times \mathsf{N}_K$, where $N_k \in \mathbf{N}_k$, for $1 \leq k \leq K$. Each of these cross-products is connected as children of each sum node in each parent region of $P$. We refer to the supplement for a detailed description, including the algorithm to construct region-graphs used in this paper.

The scope function $\psi$ of the SPN is inherited from the $\psi$ of the region graph: any SPN node introduced for a region (partition) gets the same scope as the region (partition) itself. It is easy to check that, if the SPN's $\mathcal{G}$ follows $\mathcal{R}$ using above construction, any proper scope function according to Definition 4 corresponds to a proper scope function according to Definition 2.

In this paper, we consider SPN structures $(\mathcal{G}, \psi)$ following a tree-shaped region graph $(\mathcal{R}, \psi)$, i.e. each node in $\mathcal{R}$ has at most one parent. Note that $\mathcal{G}$ is in general *not* tree-shaped in this case, unless $I = J = 1$. Further note, that this sub-class of SPNs is still very expressive, and that many SPN learners, e.g. [10, 27], also restrict to it.

When the SPN follows a tree-shaped region graph, the scope function can be encoded as follows. Let $P$ be any partition and $R_1, \ldots, R_{|\mathbf{ch}(P)|}$ be its children. For each data dimension $d$, we introduce a discrete latent variable $Y_{P,d}$ with $1, \ldots, |\mathbf{ch}(P)|$ states. Intuitively, the latent variable $Y_{P,d}$ represents a decision to assign dimension $d$ to a particular child, given that all partitions "above" have decided to assign $d$ onto the path leading to $P$ (this path is unique since $\mathcal{R}$ is a tree). More formally, we define:

**Definition 5** (Induced scope function). *Let* $\mathcal{R}$ *be a tree-shaped region graph structure, let* $Y_{P,d}$ *be defined as above, let* $\mathbf{Y} = \{Y_{P,d}\}_{P \in \mathcal{R}, d \in \{1 \ldots D\}}$, *and let* $\mathbf{y}$ *be any assignment for* $\mathbf{Y}$. *Let* $Q$ *denote any node in* $\mathcal{R}$, *let* $\Pi$ *be the unique path from the root to* $Q$ *(exclusive* $Q$*). The scope function induced by* $\mathbf{y}$ *is defined as:*

$$\psi_{\mathbf{y}}(Q) := \left\{ X_d \,\middle|\, \prod_{P \in \Pi} \mathbb{1}[R_{y_{P,d}} \in \Pi] = 1 \right\}, \tag{5}$$

*i.e.* $\psi_{\mathbf{y}}(Q)$ *contains* $\mathbf{X}_d$ *if for each partition in* $\Pi$ *also the child indicated by* $y_{P,d}$ *is in* $\Pi$.

It is easy to check that for any tree-shaped $\mathcal{R}$ and any $\mathbf{y}$, the induced scope function $\psi_{\mathbf{y}}$ is a proper scope function according to Definition 4. Conversely, for any proper scope function according to Definition 4, there exists a $\mathbf{y}$ such that $\psi_{\mathbf{y}} \equiv \psi$.[2]

We can now incorporate $\mathbf{Y}$ in our model. Therefore, we assume Dirichlet priors for each $Y_{P,d}$ and extend the generative model (4) as follows:

$$
\begin{aligned}
\mathbf{w}_{\mathsf{S}} \mid \alpha &\sim \mathcal{D}ir(\mathbf{w}_{\mathsf{S}} \mid \alpha) \ \ \forall \mathsf{S}\,, & z_{\mathsf{S},n} \mid \mathbf{w}_{\mathsf{S}} &\sim \mathcal{C}at(z_{\mathsf{S},n} \mid \mathbf{w}_{\mathsf{S}}) \ \ \forall \mathsf{S} \, \forall n, \\
\mathbf{v}_P \mid \beta &\sim \mathcal{D}ir(\mathbf{v}_P \mid \beta) \ \ \forall P\,, & y_{P,d} \mid \mathbf{v}_P &\sim \mathcal{C}at(v_{P,d} \mid \mathbf{v}_P) \ \ \forall P \, \forall d, \\
\theta_{\mathsf{L}} \mid \gamma &\sim p(\theta_{\mathsf{L}} \mid \gamma) \ \ \forall \mathsf{L}\,, & \mathbf{x}_n \mid \mathbf{z}_n, \mathbf{y}, \theta &\sim \prod_{\mathsf{L} \in T(\mathbf{z}_n)} \mathsf{L}(\mathbf{x}_{\mathbf{y},n} \mid \theta_{\mathsf{L}}) \ \ \forall n.
\end{aligned}
\tag{6}
$$

Here, the notation $\mathbf{x}_{\mathbf{y},n}$ denotes the evaluation of $\mathsf{L}$ on the scope induced by $\mathbf{y}$. Figure 2 illustrates our generative model in plate notation, in which directed edges indicate dependencies between variables. Furthermore, our Bayesian formulation naturally allows for various nonparametric formulations of SPNs. In particular, one can use the stick-breaking construction [36] of a Dirichlet process mixture model with SPNs as mixture components. We illustrate this approach in the experiments.[3]

## 5   Sampling-based Inference

Let $\mathcal{X} = \{\mathbf{x}_n\}_{n=1}^N$ be a training set of $N$ observations $\mathbf{x}_n$, we aim to draw posterior samples from our generative model given $\mathcal{X}$. For this purpose, we perform Gibbs sampling alternating between i) updating parameters $\mathbf{w}, \theta$ (fixed $\mathbf{y}$), and ii) updating $\mathbf{y}$ (fixed $\mathbf{w}, \theta$).

**Updating Parameters $\mathbf{w}, \theta$ (fixed $\mathbf{y}$)**   We follow the same procedure as in [43], i.e. in order to sample $\mathbf{w}$ and $\theta$, we first sample assignments $\mathbf{z}_n$ for all the sum latent variables $\mathbf{Z}_n$ in the SPN, and subsequently sample new $\mathbf{w}$ and $\theta$. For a given set of parameters ($\mathbf{w}$ and $\theta$), each $\mathbf{z}_n$ can be drawn independently and follows standard SPN ancestral sampling. The latent variables not visited during ancestral sampling, are drawn from the prior. After sampling all $\mathbf{z}_n$, the sum-weights are sampled from the posterior distributions of a Dirichlet, i.e. $\mathcal{D}ir(\alpha + c_{\mathsf{S},1}, \dots, \alpha + c_{\mathsf{S},K_{\mathsf{S}}})$ where $c_{\mathsf{S},k} = \sum_{n=1}^N \mathbb{1}[z_{\mathsf{S},n} = k]$ denotes the number of observations assigned to child $k$. The parameters at leaf nodes can be updated similarly; see [43] for further details.

**Updating the Structure $\mathbf{y}$, (fixed $\mathbf{w}, \theta$)**   We use a collapsed Gibbs sampler to sample all $y_{P,d}$ assignments. For this purpose, we marginalise out $\mathbf{v}$ (c.f. the dependency structure in Figure 2) and sample $y_{P,d}$ from the respective conditional. Therefore, let $\mathbf{y}_P$ denote the set of all dimension assignments at partition $P$ and let $\mathbf{y}_{P,\not{d}}$ denote the exclusion of $d$ from $\mathbf{y}_P$. Further, let $\mathbf{y}_{\mathbf{P} \setminus P, d}$ denote the assignments of dimension $d$ on all partitions except partition $P$, then the conditional probability of assigning dimension $d$ to child $k$ under $P$ is:

$$p(y_{P,d} = k \mid \mathbf{y}_{P,\not{d}}, \mathbf{y}_{\mathbf{P} \setminus P,d}, \mathcal{X}, \mathbf{z}, \theta, \beta) = p(y_{P,d} = k \mid \mathbf{y}_{P,\not{d}}, \beta) p(\mathcal{X} \mid y_{P,d} = k, \mathbf{y}_{\mathbf{P} \setminus P,d}, \mathbf{z}, \theta). \quad (7)$$

Note that the conditional prior in Equation 7 follows standard derivations, i.e. $p(y_{P,d} = k \mid \mathbf{y}_{P,\not{d}}, \beta) = \frac{\beta + m_{P,k}}{\sum_{j=1}^{|\mathbf{ch}(P)|} \beta + m_{P,k}}$ , where $m_{P,k} = \sum_{d \in \psi(P) \setminus d} \mathbb{1}[y_{P,d} = k]$ are component counts. The second term in Equation 7 is the product over marginal likelihood terms of each product node in $P$. Intuitively, values for $y_{P,d}$ are more likely if other dimensions are assigned to the same child (rich-get-richer) and if the product of marginal likelihoods of child $k$ has low variance in $d$.

Given a set of $T$ posterior samples, we can compute predictions for an unseen datum $\mathbf{x}^*$ using an approximation of the posterior predictive distribution, i.e.

$$p(\mathbf{x}^* \mid \mathcal{X}) \approx \frac{1}{T} \sum_{t=1}^T \mathcal{S}(\mathbf{x}^* \mid \mathcal{G}, \psi_{\mathbf{y}^{(t)}}, \mathbf{w}^{(t)}, \theta^{(t)}), \quad (8)$$

where $\mathcal{S}(\mathbf{x}^* \mid \mathcal{G}, \psi_{\mathbf{y}^{(t)}}, \mathbf{w}^{(t)}, \theta^{(t)})$ denotes the SPN of the $t^{\text{th}}$ posterior sample with $t = 1, \dots, T$. Note that we can represent the resulting distribution as a single SPN with $T$ children (sub-SPNs).

## 6   Experiments

We assessed the performance of our approach on discrete [10] and heterogeneous data [43] as well as on three datasets with missing values. We constructed $\mathcal{G}$ using the algorithm described in the supplement and used a grid search over the parameters of the graph. Further, we used $5 \cdot 10^3$ burn-in steps and estimated the predictive distribution using $10^4$ samples from the posterior. Since the Bayesian framework is protected against overfitting, we combined training and validation sets and followed classical Bayesian model selection [34], i.e. using the Bayesian model evidence. Note that within the validation loop, the computational graph remains fixed. We list details on the selected

Table 1: Average test log-likelihoods on discrete datasets using SOTA, Bayesian SPNs (ours) and infinite mixtures of SPNs (ours$^\infty$). Significant differences are underlined. Overall best result is in bold. In addition we list the best-to-date (BTD) results obtained using SPNs, PSDDs or CNets.

| Dataset | LearnSPN | RAT-SPN | CCCP | ID-SPN | ours | ours$^\infty$ | BTD |
|---|---|---|---|---|---|---|---|
| NLTCS | −6.11 | −6.01 | −6.03 | −6.02 | **−6.00** | −6.02 | −5.97 |
| MSNBC | −6.11 | −6.04 | −6.05 | −6.04 | −6.06 | **−6.03** | −6.03 |
| KDD | −2.18 | −2.13 | −2.13 | −2.13 | **−2.12** | −2.13 | −2.11 |
| Plants | −12.98 | −13.44 | −12.87 | **−12.54** | −12.68 | −12.94 | −11.84 |
| Audio | −40.50 | −39.96 | −40.02 | −39.79 | **−39.77** | −39.79 | −39.39 |
| Jester | −53.48 | −52.97 | −52.88 | −52.86 | −52.42 | −52.86 | −51.29 |
| Netflix | −57.33 | −56.85 | −56.78 | −56.36 | −56.31 | −56.80 | −55.71 |
| Accidents | −30.04 | −35.49 | −27.70 | **−26.98** | −34.10 | −33.89 | −26.98 |
| Retail | −11.04 | −10.91 | −10.92 | −10.85 | −10.83 | **−10.83** | −10.72 |
| Pumsb-star | −24.78 | −32.53 | −24.23 | **−22.41** | −31.34 | −31.96 | −22.41 |
| DNA | −82.52 | −97.23 | −84.92 | **−81.21** | −92.95 | −92.84 | −81.07 |
| Kosarak | −10.99 | −10.89 | −10.88 | **−10.60** | −10.74 | −10.77 | −10.52 |
| MSWeb | −10.25 | −10.12 | −9.97 | **−9.73** | −9.88 | −9.89 | −9.62 |
| Book | −35.89 | −34.68 | −35.01 | −34.14 | **−34.13** | −34.34 | −34.14 |
| EachMovie | −52.49 | −53.63 | −52.56 | −51.51 | −51.66 | **−50.94** | −50.34 |
| WebKB | −158.20 | −157.53 | −157.49 | **−151.84** | −156.02 | −157.33 | −149.20 |
| Reuters-52 | −85.07 | −87.37 | −84.63 | **−83.35** | −84.31 | −84.44 | −81.87 |
| 20 Newsgrp | −155.93 | −152.06 | −153.21 | **−151.47** | −151.99 | −151.95 | −151.02 |
| BBC | −250.69 | −252.14 | **−248.60** | −248.93 | −249.70 | −254.69 | −229.21 |
| AD | −19.73 | −48.47 | −27.20 | **−19.05** | −63.80 | −63.80 | −14.00 |

parameters and the runtime for each dataset in the supplement, c.f. Table 3. For posterior inference in infinite mixtures of SPNs, we used the distributed slice sampler [8].

Table 1 lists the test log-likelihood scores of state-of-the-art (SOTA) structure learners, i.e. LearnSPN [10], LearnSPN with parameter optimisation (CCCP) [46] and ID-SPN [35], random region-graphs (RAT-SPN) [27] and the results obtained using Bayesian SPNs (ours) and infinite mixtures of Bayesian SPN (ours$^\infty$) on discrete datasets. In addition we list the best-to-date (BTD) results, collected based on the most recent works on structure learning for SPNs [12], PSDDs [19] and CNets[21, 30]. Note that the BTD results are often by large ensembles over structures. Significant differences to the best SOTA approach under the Mann-Whitney-U-Test [20] with $p < 0.01$ are underlined. We refer to the supplement for an extended results table and further details on the significance tests. We see that Bayesian SPNs and infinite mixtures generally improve over LearnSPN and RAT-SPN. Further, in many cases, we observe an improvement over LearnSPN with additional parameter learning and often obtain results comparable to ID-SPN or sometimes outperforms BTD results. Note that ID-SPN uses a more expressive SPN formulation with Markov networks as leaves and also uses a sophisticated learning algorithm.

Additionally, we conducted experiments on heterogeneous data, see: [22, 43], and compared against mixed SPNs (MSPN) [22] and ABDA [43]. We used mixtures over likelihood models as leaves, similar to Vergari *et al.* [43], and performed inference over the structure, parameters and likelihood models. Further details can be found in the supplement. Table 2 lists the test log-likelihood scores of all approaches, indicating that our approaches perform comparably to structure learners tailored to heterogeneous datasets and sometimes outperform SOTA. [4] Interestingly, we obtain, with a large margin, better test scores for `Autism` which might indicate that existing approaches overfit in this case while our formulation naturally penalises complex models.

We compared the test log-likelihood of LearnSPN, ID-SPN and Bayesian SPNs against an increasing number of observations having $50\%$ dimensions missing completely at random [28]. We evaluated LearnSPN and ID-SPN by i) removing all observations with missing values and ii) using K-nearest neighbour imputation [2] (denoted with an asterisk). Note that we selected k-NN imputation because it arguably provides a stronger baseline than simple mean imputation (while being computationally

Table 2: Average test log-likelihoods on heterogeneous datasets using SOTA, Bayesian SPN (ours) and infinite mixtures of SPNs (ours$^\infty$). Overall best result is indicated in bold.

| Dataset | MSPN | ABDA | ours | ours$^\infty$ |
|---|---|---|---|---|
| Abalone | **9.73** | 2.22 | 3.92 | 3.99 |
| Adult | −44.07 | −5.91 | **−4.62** | −4.68 |
| Australian | −36.14 | **−16.44** | −21.51 | −21.99 |
| Autism | −39.20 | −27.93 | **−0.47** | −1.16 |
| Breast | −28.01 | −25.48 | **−25.02** | −25.76 |
| Chess | −13.01 | −12.30 | **−11.54** | −11.76 |
| Crx | −36.26 | **−12.82** | −19.38 | −19.62 |
| Dermatology | −27.71 | −24.98 | **−23.95** | −24.33 |
| Diabetes | −31.22 | **−17.48** | −21.21 | −21.06 |
| German | −26.05 | **−25.83** | −26.76 | −26.63 |
| Student | −30.18 | **−28.73** | −29.51 | −29.9 |
| Wine | **−0.13** | −10.12 | −8.62 | −8.65 |

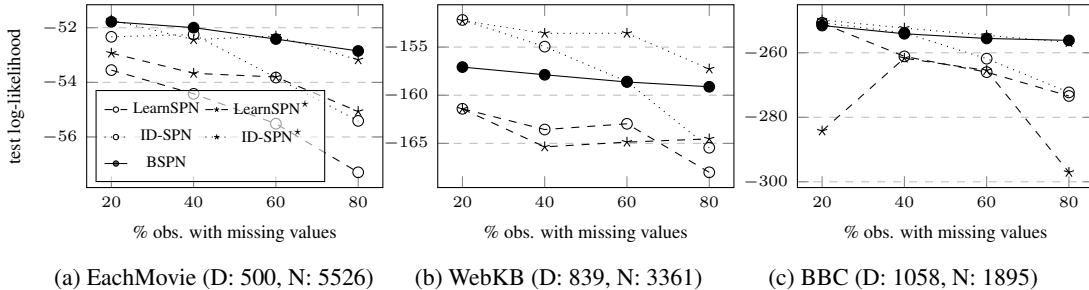

(a) EachMovie (D: 500, N: 5526)   (b) WebKB (D: 839, N: 3361)   (c) BBC (D: 1058, N: 1895)

Figure 3: Performance under missing values for discrete datasets with increasing dimensionality (D). Results for LearnSPN are shown in dashed lines, results for ID-SPN in dotted lines and our approach is indicated using solid lines. Star (⋆) indicates k-NN imputation while (∘) means no imputation.

more demanding). All methods have been trained using the full training set, i.e. training and validation set combined, and were evaluated using default parameters to ensure a fair comparison across methods and levels of missing values. See supplement Section A.3 for further details. Figure 3 shows that our formulation is consistently robust against missing values while SOTA approaches often suffer from missing values, sometimes even if additional imputation is used.

## 7 Conclusion

Structure learning is an important topic in SPNs, and many promising directions have been proposed in recent years. However, most of these approaches are based on intuition and refrain from declaring an explicit and global principle to structure learning. In this paper, our primary motivation is to *change* this practice. To this end, we phrase structure (and joint parameter) learning as Bayesian inference in a latent variable model. Our experiments show that this principled approach competes well with prior art and that we gain several benefits, such as automatic protection against overfitting, robustness under missing data and a natural extension to nonparametric formulations.

A critical insight for our approach is to decompose structure learning into two steps, namely constructing a computational graph and separately learning the SPN's scope function – determining the "effective" structure of the SPN. We believe that this novel approach will be stimulating for future work. For example, while we used Bayesian inference over the scope function, it could also be optimised, e.g. with gradient-based techniques. Further, more sophisticated approaches to identify the computational graph, e.g. using AutoML techniques or neural structural search (NAS) [47], could be fruitful directions. The Bayesian framework presented in this paper allows several natural extensions, such as parameterisations of the scope-function using hierarchical priors or using variational inference for large-scale approximate Bayesian inference, and relaxing the necessity of a given computational graph, by incorporating nonparametric priors in all stages of the model formalism.

## Acknowledgements

This work was partially funded by the Austrian Science Fund (FWF): I2706-N31 and has received funding from the European Union's Horizon 2020 research and innovation programme under the Marie Skłodowska-Curie Grant Agreement No. 797223 — HYBSPN.

## Footnotes

[1]The scope of a node is a subset of random variables, the node is responsible for and needs to fulfil the so-called completeness and decomposability conditions, see Section 3.

[2] Note that the relation between $\mathbf{y}$ and $\psi_{\mathbf{y}}$ is similar to the relation between $\mathbf{z}$ and $\mathcal{T}$, i.e. each $\psi_{\mathbf{y}}$ corresponds in general to many $\mathbf{y}$'s. Also note, the encoding of $\psi$ for general region graphs is more envolved, since the path to each $Q$ is not unique anymore, requiring consistency among $y$.

[3]See `https://github.com/trappmartin/BayesianSumProductNetworks` for and implementation of Bayesian SPNs in form of a Julia package accompanied by codes and datasets used for the experiments.

[4] Note that we did not apply a significance test as implementations of existing approaches are not available.

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
