[Supplementary Material]

# Supplement - Bayesian Learning of Sum-Product Networks

**Martin Trapp**[1,2]**, Robert Peharz**[3]**, Hong Ge**[3]**,**
**Franz Pernkopf**[1]**, Zoubin Ghahramani**[4,3]
[1]Graz University of Technology, [2]Austrian Research Institute for AI,
[3]University of Cambridge, [4]Uber AI
`martin.trapp@tugraz.at`, `rp587@cam.ac.uk`, `hg344@cam.ac.uk`
`pernkopf@tugraz.at`, `zoubin@eng.cam.ac.uk`

## A Experiments

### A.1 Setup

As described in the paper, we: 1) We combined the training and validation set to a single training set. 2) We used $5 * 10^3$ burn-in steps and estimated the training and testing performance from samples of a chain of $10^4$ samples. 3) We used a grid search over the number of nodes per region $I \in [5, 10]$, number of nodes per atomic region $I \leq J \in [5, 10]$, number of partitions under a region $M \in [2, 4, 8]$, and the depth, i.e. consecutive region-partition layers, $L \in [1, 2]$ and selected the best configuration according to the model evidence. In the experiments we used the following hyper-parameters for the symmetric Dirichlet priors: $\alpha = 1.0$ as concentration parameter for all sum nodes, $\beta = 10.0$ as concentration parameter for all product nodes to enforce partitions into equally size parts.

We ran all experiments on a high performance cluster using multi-threaded computations. The SLURM script and the necessary code and datasets to run the experiments with the respective number of threads can be found on `https://github.com/trappmartin/BayesianSumProductNetworks`.

### A.2 Heterogeneous Experiments

To conduct the heterogeneous data experiments, we introduce mixtures over likelihood model for each leaf node. In particular, we used the following likelihood and prior constructions in the experiment.

| Datatype | Likelihood | Prior |
|---|---|---|
| Continuous | Gaussian i.e., $x_d \sim \mathcal{N}(\mu, \sigma^2)$ | $\sigma^2 \sim \Gamma^{-1}(2.0, 3.0)$ |
| | | $\mu \sim \mathcal{N}(\tilde{\mu}, \sigma^2)$ |
| Continuous | Exponential i.e., $x_d \sim \mathcal{E}xp(\lambda)$ | $\lambda \sim \Gamma(1.0, 1.0)$ |
| Discrete | Poisson i.e., $x_d \sim \mathcal{P}oisson(\lambda)$ | $\lambda \sim \Gamma(1.0, 1.0)$ |
| Discrete | Categorical i.e., $x_d \sim \mathcal{C}at(w)$ | $w \sim \mathcal{D}ir(0.1)$ |
| Discrete | Bernoulli i.e., $x_d \sim \mathcal{B}ern(p)$ | $p \sim \mathcal{B}eta(0.5, 0.5)$ |

Table 1: Likelihood functions and priors used for heterogeneous data experiments.

The distribution of each leaf factorises as:

$$\mathsf{L} = \prod_{X_i \in \psi(\mathsf{L})} \sum_k w_k p(X_i \,|\, \theta_{\mathsf{L}, i, k}) \tag{1}$$

and we used a symmetric Dirichlet prior with concentration parameter $\alpha = 0.1$ for the weights of the mixture to ensure only few components are selected. This approach is similar to the model in [4].

## A.3 Missing Data Experiments

We evaluated the robustness of learnSPN, ID-SPN and Bayesian SPN against missing values in the training data. For this purpose, we artificially introduced missing values completely at random in the training and validation set of `EachMovie`, `WebKB` and `BBC`. We evaluate their performance in the cases of 20%, 40%, 60% or 80% of all observations having 50% missing values. All methods have been trained using the full training set, i.e. training and validation set combined, and where evaluated using the following default parameters: (1) LearnSPN: cluster penalty = 0.6, significance threshold = 10 as described in [1]; (2) ID-SPN: using the default settings described in [3]; and (3) Bayesian SPN: $I = 5$ nodes per region, $J = 10$ nodes per atomic region, $R = 8$ partitions under a region, and a depth of $L = 1$.

## A.4 Statistical Significance Tests

To assess the statistical significance of the reported results we computed the $p$-value of the Mann-Whitney-U-Test [2]. The Mann-Whitney-U-Test is a nonparametric equivalent of the two sample $t$-test which does not require the assumption of normal distributions. The respective $p$-values obtained from the Mann-Whitney-U-Test for Bayesian SPNs and infinite mixtures of Bayesian SPNs are listed in Table 2.

Table 2: Mann-Whitney-U-Test $p$-values of Bayesian SPNs (a) and infinite mixtures of Bayesian SPNs (b) compared with LearnSPN, RAT-SPN and ID-SPN. Values below the $0.01$ threshold are underlined.

|  | (a) Bayesian SPNs. | | | (b) Infinite mixtures of Bayesian SPNs. | | |
|---|---|---|---|---|---|---|
| Dataset | LearnSPN | RAT-SPN | ID-SPN | LearnSPN | RAT-SPN | ID-SPN |
| NLTCS | 0.726 | 0.573 | 0.291 | 0.887 | 0.950 | 0.123 |
| MSNBC | 0.634 | 0.420 | 0.474 | 0.911 | 0.173 | 0.842 |
| KDD | 0.792 | 0.044 | 0.505 | 0.755 | 0.050 | 0.472 |
| Plants | $< 0.001$ | $< 0.001$ | $< 0.001$ | $< 0.001$ | $< 0.001$ | $< 0.001$ |
| Audio | $< 0.001$ | $< 0.001$ | $< 0.001$ | $< 0.001$ | $< 0.001$ | $< 0.001$ |
| Jester | $< 0.001$ | $< 0.001$ | 0.908 | 0.004 | 0.885 | 0.001 |
| Netflix | 0.100 | 0.924 | 0.455 | 0.107 | 0.944 | 0.442 |
| Accidents | $< 0.001$ | $< 0.001$ | $< 0.001$ | $< 0.001$ | $< 0.001$ | $< 0.001$ |
| Retail | $< 0.001$ | 0.002 | 0.023 | $< 0.001$ | 0.008 | 0.020 |
| Pumsb-star | $< 0.001$ | $< 0.001$ | $< 0.001$ | $< 0.001$ | 0.025 | $< 0.001$ |
| DNA | $< 0.001$ | 0.001 | 0.084 | $< 0.001$ | $< 0.001$ | 0.023 |
| Kosarak | $< 0.001$ | $< 0.001$ | $< 0.001$ | $< 0.001$ | $< 0.001$ | $< 0.001$ |
| MSWeb | 0.721 | 0.005 | 0.030 | 0.354 | 0.047 | 0.002 |
| Book | $< 0.001$ | 0.035 | 0.124 | 0.034 | 0.845 | 0.442 |
| EachMovie | 0.270 | 0.228 | 0.390 | 0.275 | 0.242 | 0.411 |
| WebKB | $< 0.001$ | 0.001 | $< 0.001$ | $< 0.001$ | $< 0.001$ | 0.645 |
| Reuters-52 | 0.089 | 0.703 | 0.998 | 0.079 | 0.638 | 0.904 |
| 20 Newsgrp | 0.846 | 0.508 | 0.969 | 0.326 | 0.636 | 0.214 |
| BBC | 0.002 | 0.288 | 0.866 | 0.002 | 0.335 | 0.795 |
| AD | 0.004 | 0.635 | 0.774 | 0.004 | 0.097 | 0.769 |

## A.5 Reported Configurations and Respective Runtimes

In addition, we computed the average runtime for a single MCMC iteration measures for an i7-6900k CPU @ 3.2 GHz. The respective runtimes for each dataset, measures for the computational graph used to report the results in the paper, are listed in Table 3. Note that these timings vary depending on the dataset size, the number of dimensions and the complexity of the computational graph.

Table 3: Number of sum nodes per region ($I$), number of leaves per atomic region ($J$), number of partitions ($M$), number of layers ($L$) and runtime (in seconds) for and iteration of MCMC sampling.

| Dataset | runtime | $I$ | $J$ | $M$ | $L$ |
|---|---|---|---|---|---|
| NLTCS | 4.03 | 5 | 10 | 8 | 2 |
| MSNBC | 33.87 | 5 | 5 | 4 | 4 |
| KDD | 43.05 | 5 | 10 | 8 | 2 |
| Plants | 24.39 | 5 | 10 | 8 | 4 |
| Audio | 3.50 | 5 | 5 | 4 | 4 |
| Jester | 5.12 | 5 | 10 | 4 | 4 |
| Netflix | 7.53 | 5 | 10 | 4 | 4 |
| Accidents | 27.55 | 10 | 10 | 8 | 4 |
| Retail | 3.46 | 10 | 10 | 4 | 2 |
| Pumsb-star | 4.15 | 10 | 10 | 8 | 2 |
| DNA | 7.92 | 5 | 10 | 8 | 4 |
| Kosarak | 10.43 | 10 | 10 | 8 | 2 |
| MSWeb | 4.91 | 5 | 5 | 8 | 2 |
| Book | 5.23 | 10 | 10 | 8 | 2 |
| EachMovie | 30.23 | 5 | 10 | 8 | 4 |
| WebKB | 3.61 | 10 | 10 | 8 | 2 |
| Reuters-52 | 6.37 | 10 | 10 | 8 | 2 |
| 20 Newsgrp | 11.02 | 5 | 10 | 8 | 2 |
| BBC | 3.56 | 5 | 10 | 8 | 2 |
| AD | 1.05 | 5 | 5 | 2 | 2 |

## A.6 Extended Results Table

In addition to the results table listed in the paper, we compare in the following table against the best-to-date results for PSDDs (btd-PDSS), the best-to-date results for CNets (btd-CNet) and the best-to-date results for SPNs (btd-SPN). Note that the best results are often obtained by very large ensembles of structure learners. Table 4 lists all results.

Table 4: Average test log-likelihoods on discrete datasets using SOTA, Bayesian SPNs (ours) and infinite mixtures of SPNs (ours$^\infty$). Overall best result is in bold.

| Dataset | LearnSPN | RAT-SPN | CCCP | ID-SPN | btd-PSDD | btd-CNet | btd-SPN | ours | ours$^\infty$ |
|---|---|---|---|---|---|---|---|---|---|
| NLTCS | −6.11 | −6.01 | −6.03 | −6.02 | −6.03 | **−5.97** | −6.01 | −6.00 | −6.02 |
| MSNBC | −6.11 | −6.04 | −6.05 | −6.04 | −6.04 | **−6.03** | −6.04 | −6.06 | **−6.03** |
| KDD | −2.18 | −2.13 | −2.13 | −2.13 | **−2.12** | −2.13 | **−2.12** | **−2.12** | −2.13 |
| Plants | −12.98 | −13.44 | −12.87 | −12.54 | −13.79 | **−11.84** | −12.54 | −12.68 | −12.94 |
| Audio | −40.50 | −39.96 | −40.02 | −39.79 | −41.98 | **−39.77** | −39.79 | **−39.77** | −39.79 |
| Jester | −53.48 | −52.97 | −52.88 | −52.86 | −53.47 | **−52.21** | −52.80 | −52.42 | −52.86 |
| Netflix | −57.33 | −56.85 | −56.78 | −56.36 | −58.41 | **−55.93** | −56.36 | −56.31 | −56.80 |
| Accidents | −30.04 | −35.49 | −27.70 | **−26.98** | −33.64 | −29.27 | −27.70 | −34.10 | −33.89 |
| Retail | −11.04 | −10.91 | −10.92 | −10.85 | **−10.81** | **−10.81** | −10.85 | −10.83 | −10.83 |
| Pumsb-star | −24.78 | −32.53 | −24.23 | **−22.41** | −33.67 | −23.44 | **−22.41** | −31.34 | −31.96 |
| DNA | −82.52 | −97.23 | −84.92 | −81.21 | −89.11 | **−81.07** | −81.21 | −92.95 | −92.84 |
| Kosarak | −10.99 | −10.89 | −10.88 | **−10.60** | −10.81 | **−10.60** | **−10.60** | −10.74 | −10.77 |
| MSWeb | −10.25 | −10.12 | −9.97 | −9.73 | −9.97 | **−9.62** | −9.73 | −9.88 | −9.89 |
| Book | −35.89 | −34.68 | −35.01 | −34.14 | −34.97 | −34.46 | −34.14 | **−34.13** | −34.34 |
| EachMovie | −52.49 | −53.63 | −52.56 | −51.51 | −56.43 | **−50.34** | −51.49 | −51.66 | −50.94 |
| WebKB | −158.20 | −157.53 | −157.49 | −151.84 | −161.09 | **−149.20** | −151.84 | −156.02 | −157.33 |
| Reuters-52 | −85.07 | −87.37 | −84.63 | −83.35 | −89.61 | **−81.87** | −83.35 | −84.31 | −84.44 |
| 20 Newsgrp | −155.93 | −152.06 | −153.21 | −151.47 | −161.09 | **−151.02** | −151.47 | −151.99 | −151.95 |
| BBC | −250.69 | −252.14 | −248.60 | −248.93 | −253.19 | **−229.21** | −248.50 | −249.07 | −254.69 |
| AD | −19.73 | −48.47 | −27.20 | −19.05 | −30.49 | **−14.00** | −19.05 | −63.80 | −63.80 |

# B  Computational Graph Construction

This section describes the algorithm to construct a computational graph, represented by a tree-shaped region-graph, as used in the paper. Note that we only consider partitions into two disjoint sub-regions. Our algorithm can, however, easily be extended for more general situations.

---

**Algorithm 1** Construction of a Computational Graph

---

**Input:** Dimensionality of dataset $D$, Number of nodes per region $I$, Number of nodes per atomic region $J$, Number of partitions under a region $M$ and depth $L$.

**function** BUILDATOMICREGION($D$, $J$)
    $R \leftarrow$ empty atomic region.
    **for** $k = 1, \ldots, J$ **do**            ▷ Equip $R$ with $J$ distribution nodes, each factorising $D$.
        Equip $R$ with $\prod_{d=1}^{D} p(\mathbf{x}| \theta_{\mathsf{L}_k,d})$.
    **end for**
    **return** $R$
**end function**

**function** BUILDREGION($D$, $I$, $J$, $M$, $L$, $l$)
    $R \leftarrow$ empty region.
    **for** $j = 1, \ldots, M$ **do**
        $P \leftarrow$ BUILDPARTITION($D$, $I$, $J$, $M$, $L$, $l + 1$)
        Make $P$ a child of $R$.
    **end for**
    Let $\mathbf{N}$ be all product nodes of all $P \in \mathbf{ch}(R)$.
    **for** $k = 1, \ldots, I$ **do**
        Equip $R$ with $\mathsf{S} = \sum_{\mathsf{P} \in \mathbf{N}} w_{\mathsf{S},\mathsf{P}} p_{\mathsf{P}}(\mathbf{x})$.
    **end for**
    **return** $R$
**end function**

**function** BUILDPARTITION($D$, $I$, $J$, $M$, $L$, $l$)
    $P \leftarrow$ empty partition.
    **if** $l = L$ **then**
        $R_1 \leftarrow$ BUILDATOMICREGION($D$, $J$)
        $R_2 \leftarrow$ BUILDATOMICREGION($D$, $J$)
    **else**
        $R_1 \leftarrow$ BUILDREGION($D$, $I$, $J$, $M$, $L$, $l$)
        $R_2 \leftarrow$ BUILDREGION($D$, $I$, $J$, $M$, $L$, $l$)
    **end if**
    Make $R_1$ and $R_2$ children of $P$.
    Let $\mathbf{N}_R$ be all nodes of $R$.
    **for** $\mathsf{N}_1 \in \mathbf{N}_{R_1}, \mathsf{N}_2 \in \mathbf{N}_{R_2}$ **do**
        Equip $P$ with $\mathsf{P} = \mathsf{N}_1 \times \mathsf{N}_2$.
    **end for**
    **return** $P$
**end function**

    **return** BUILDREGION($D$, $I$, $J$, $M$, $L$, 0)

---

Once constructed, the scope function of a computational graph defined as a region graph can be inferred using posterior inference. Given a proper scope function, we can obtain the effective SPN structure by simply removing each sub-tree with empty scope. If necessary, one can further extracting the network located at the root (sum) node by traversing down the nodes in the region graph, c.f. Algorithm 1. Note that it is not necessary to extract the network structure from the region graph as one can directly work with the effective SPN represented by the region graph.