[Reviews · NeurIPS 2019]

Reviewer 1



The paper is technically sound, clearly written and leads the reader through the steps required to understand the formulations. The contribution is significant as it solves a difficult task previously avoided in the literature. The empirical section places the method as a strong contender in the learning stack of SPNs. A task that is increasingly difficult as there is constant progress evaluated on the same benchmarks. The contribution is original, the authors give clear references to previous work that is related to bayesian learning and region graph structure on SPNs.

Reviewer 2



**************** Thanks for the detailed response, which addressed most of my concerns (based on some promise). Given the space constraint of the rebuttal, I will trust the authors to indeed incorporate the changes as promised, and given this I increased my score. ***************** This paper proposes a novel method to do structure learning for SPNs. However, at several places in this paper, it is too dense to follow. More detailed comments are as follows. First, this paper lacks a dedicated related work section. There is some brief discussion about how this work differs from existing literature, in the introduction, yet it is not enough. Section 2 is called "Background & Related works". However, I would say 95% of this section is just background knowledge on SPNs. More importantly, this paper lacks coverage about related works that are not about SPNs but on some other dialects of tractable graphical models (for example, cutset networks and probabilistic sentential decision diagrams, etc.) Second, it is not easy to follow exactly how a computation graph is constructed from a region graph. Here, having an example figure would be of great help. Furthermore, it seems Figure 1 never gets mentioned in the paper. And I am also not sure how to exactly interpret Figure 1 given no explanations accompany it. Third, this point is related to my first point. In experiments, I also encourage the authors to include results from structure learning methods on other dialects of tractable graphical models to form a real SOTA baselines. Also I encourage the authors to drop the arrows in the tables. They dilute the focus of the tables, which should be the bold/underlined numbers. Fourth, I am not quite convinced by the results on missing data. Mean imputation, the arguably most popular imputation method, is missing from the results. More importantly, I believe the baselines may be too weak. One natural way to deal with missing data is to run EM + MPE inferences. Fifth, in some sense, I am not quite convinced that the method proposed in this paper is a complete structure learning algorithm, as the region graph and hence the computation graph is predefined. In other words, this paper’s structure learning is solely on learning the scope functions. Perhaps learning the scope functions alone is enough? If so, can the authors provide a compelling for it? Typos: parametrization -> parameterization on line 13, 61, 74,…

Reviewer 3



This paper proposes a Bayesian approach to learn a SPN model from data. Unlike the existing methods that jointly estimate the computational graph and scope function, the paper separates the two tasks, and claims that the first task is similar to neural network structure validation. The paper assumes the graph is given, then parameterizes the scope function and SPN distribution so that a Bayesian generative model can both sample SPN parameters and determine the scope function. Then a Gibbs sampling algorithm is developed for the posterior inference, which in turn result in the learned SPN model and parameters. The experiments verified the effectiveness of the proposed approach. Overall, this is a piece of interesting and solid work. The ideas of using induced trees to parameterize SPN distribution and using regional graphs to parameterize the scope function are amazing, because they render a Bayesian model formulation that can include both the SPN distribution parameters and the scope function. My major concern is another task --- the identification of the computational graph. Although it is reasonable to separate the two tasks, the paper seems to overlook the difficulty or importance of the structure identification. In fact, identifying appropriate NN structure is non-trivial and require massive computational resources. The authors are referred to the recent work of AutoML. Listed are my detailed concerns and suggestions. (1) Although the authors claim that they decompose the SPN model learning into the two tasks, they never give a detailed solution of the first task, i.e., computational graph identification. They always assume the graph is given. A natural problem is, once the scope function is learned for a particular graph, what is the next step? Do we have an iterative procedure to switch to a better graph, and then learn the scope function again? Although the first step is not the major focus, the paper should at least discuss possible methods. (2) Since there have already been prior work that jointly learn the graph and scope function, it seems unfair to fix the graph structure during the evaluation. The authors probably can embed the scope function learning approach into an existing framework that can search for graphs and then compare with the existing approaches. Then the comparison results will be much more convincing.

[Author Response · NeurIPS 2019]



Figure 1: An example of a computational graph $\mathcal{G}$ (left) and a sum-product-network (SPN) structure (right), defined by the scope function $\psi$, discovered using posterior inference on **y**. The resulting SPN might contain only a subset of the nodes in $\mathcal{G}$ as some sub-trees might be allocated with an empty scope during inference (dotted) – evaluating to constant 1. The graph $\mathcal{G}$ only encodes the topological layout of nodes, while the "effective" SPN structure is encoded via $\psi$. Example will be included in the supplementary.

**General remarks**   **We want to thank all reviewers for their constructive feedback and for reviewing our work.**
Fig. 1 depicts our proposed decomposition into computational graph and scope function. See caption for details.

**Reviewer 1**   **Running times for Gibbs sampling:** We will report detailed running times in the revised paper. For
now, we report total running times on the cross-validated computational graphs, for a diverse selection of datasets.
*Audio* ($N = 17000$, $D = 100$): 13h, 17m 49s; *EachMovie* ($N = 5526$, $D = 500$): 106h 54m 04s; *BBC* ($N = 1895$,
$D = 1085$): 12h 49m 45s. These times were measured for an i7-6900k CPU @ 3.2 GHz.

**Reviewer 2**   **Related work:** We will restructure Sections 1 & 2 to provide a related work section, and incorporate
related tractable probabilistic models. In particular, we will add (i) (extremely randomized) cutset networks [Rahman et
al. 2014, Di Mauro et al. 2017], (ii) probabilistic sentential decision diagrams [Kisa et al. 2014, Liang et al. 2017] and
(iii) mixtures of trees [Meila 2000].

**Region-graph and Fig. 1:** Section 3.2 contains a brief description of how to construct computational graphs from
region graphs, which is admittedly quite terse. We will augment this description with a detailed description in the
supplementary. Reference to Fig. 1 was accidentally removed in one of our paper iterations. This will be fixed in the
revised paper.

**SOTA results:** We will add results of other tractable probabilistic models into the results table. In particular, we will list
the latest results reported for: (i) cutset networks, (ii) probabilistic sentential decision diagrams and (iii) sum-product
networks. Also, we will add a column listing the best results (considering all published results on tractable probabilistic
models) for each dataset and drop the arrows in the tables.

**Missing values:** We selected k-NN imputation because it arguably provides a stronger baseline than simple mean
imputation (while being computationally more demanding). Pairing structure learning with EM + MPE would be
a possible avenue. However, using EM as an inner loop within a structure search would be computationally quite
demanding. Using (approximate) MPE inference within a structure search is heuristic. Our Bayesian SPN framework
is, as far as we know, the first method that allows coherent structure learning under missing data.

**Learning only scope functions:** We indeed focus on learning the scope function, as it is clearly the more challenging
part – the computational graph has only the requirement to be acyclic, which could be addressed with tools from neural
architecture search, AutoML, or, in future work, with Bayesian inference over graphs using more flexible priors.

**Reviewer 3**   **Computational graph identification:** We indeed decompose the sum-product network (SPN) structure
learning problem into two parts, namely (i) determining a computational graph and (ii) learning the scope function. In
our paper, we emphasise the latter aspect as it is far more challenging, due to SPNs' structural constraints – completeness
and decomposability. Determining the computational graph is far simpler, and can be tackled with cross-validation
(as in this paper), or as suggested by the reviewer using AutoML techniques or neural structural search (NAS). The
reviewer is right that these directions are natural, but we leave them to future work.

**Fixing computational structure:** We do not have an iterative procedure to switch to a better computational graph, we
only perform cross-validation over 24 different computational graphs. Within the validation loop, the computational
graph $\mathcal{G}$ remains fixed. Fixing the computational graph structure is only unfair towards our approach.

**Existing structure learners:** Embedding our approach into existing structure learners is non-trivial, as all existing
methods learn the computational graph and the scope function in an entangled way. As our paper focuses on introducing
a new way of thinking about structure learning and stimulating research on Bayesian formulations, we leave those
directions to future work.

[Meta-Review · NeurIPS 2019]

This paper proposes a Bayesian formulation for SPN models. All reviewers felt this formulation had merit, and there was agreement with the authors on possible improvments.